# A Magnetic Geothermometer in Moderately Buried Shales

**Charles Aubourg [1],*, Myriam Kars [2], Jean-Pierre Pozzi [3], Martin Mazurek [4] and Olivier Grauby [5]**

1   Laboratoire des Fluides Complexes et leurs Réservoirs, UMR 5150 CNRS TOTAL,
    Université de Pau et des Pays de l'Adour, Avenue de l'Université, CEDEX, 64013 Pau, France
2   Center for Advanced Marine Core Research, Kochi University, B200 Monobe, Nankoku 783-8502, Japan;
    mkars@kochi-u.ac.jp
3   Laboratoire de Géologie, CNRS-Ecole Normale Supérieure PSL University, 75013 Paris, France;
    pozzi@biotite.ens.fr
4   Rock-Water Interaction, Institute of Geological Sciences, University of Bern, Baltzerstrasse 3,
    3012 Bern, Switzerland; martin.mazurek@geo.unibe.ch
5   Cinam, UMR 7325 CNRS, Campus de Luminy, Aix-Marseille Université, CEDEX 9, 13288 Marseille, France;
    olivier.grauby@univ-amu.fr
*   Correspondence: charles.aubourg@univ-pau.fr

**Abstract:** Shales contain magnetic minerals generally at very low concentrations. In the early stages of diagenesis, the inherited magnetic minerals are altered, while magnetic nanominerals are formed. In this study, we proposed a study of shales over a stratigraphic thickness of 1.3 km from a borehole in the Paris basin (Borehole EST 433, France), and shales from the same formation (Opalinus Clay) collected in seven boreholes in the Jura molasse basin (Swiss). Magnetic measurements at experimental temperatures <30 K allowed the formation of a proxy of magnetite nanograins named PM. We showed that some of these nanograins formed around the pyrite grains, probably under the action of temperature and organic matter. PM was then compared to the maturity values of the organic matter. We found a correlation between PM and the percentage of reflectance of vitrinite. The shales from both Paris and molassic Swiss basins showed very comparable magnetic characteristics for a given maturity level. The magnetic study therefore provided constraints on the maturity level of the shales in the oil window area. Our study showed that PM can be used as a geothermometer in shales in which $CaCO_3$ is lower than 60%.

**Keywords:** shales; magnetite; goethite; oil window; geothermometer

## 1. Introduction

Shales, which account for two-thirds of the volume of sedimentary rocks [1], are of significant economic interest for geological storage, unconventional resources, and fluid transfers. The diagenesis of shales during burial modifies their petrophysical properties [2], and it is therefore relevant to evaluate the maximum burial conditions in shales. Many geothermometers are used to determine the peak temperature of burials. For burial temperatures below 150 °C, common geothermometers in shales include RockEval pyrolysis [3], vitrinite reflectance [4], fluid inclusions [5], illite crystallinity [6], and apatite fission tracks [7]. In the Paris basin (France), Blaise et al [8] proposed a comparative study of several techniques, including an original study based on the magnetic properties of shales, initially proposed by Aubourg and Pozzi [9]. In this study, we propose to compare the use of the geothermometer according to two approaches; one consisting of describing, in a more complete way than in the study of Blaise et al. [8], the evolution of the magnetic parameters through different sedimentary formations (Drilling EST433); and the other consisting of studying the same formation (Opalinus Clay) at different burial depths in the Swiss Molasse Basin.

## 2. Rock Magnetism of Shales

Aubourg and Pozzi [9] initially suggested that the magnetic minerals of argillaceous rocks may be representative of modest burial; i.e., less than 10 km (assuming a thermal gradient of 25 °C/km). By analyzing the magnetic properties of shales at low analytical temperature, in the range of 10 K to 300 K (−263 °C to 27 °C), they reported a diagnostic signature, from which a PM parameter could be extracted (Figure 1A). When a strong magnetic field (typically on the order of 2.5 Tesla) is applied to a rock, an isothermal remanent magnetization will be recorded in the ferromagnetic minerals. The remanent magnetization is named the saturated isothermal remanent magnetization (SIRM). If this magnetic field is applied at room temperature (RT = 300 K), a certain fraction of magnetic minerals will record this magnetic field. This magnetization is called an RT-SIRM. This fraction includes single-domain (SD) and multidomain (MD) magnetic grains. For example, for magnetite, these are grains with a size greater than 20 nm [10]. If the magnetic field is applied to the rock at a very low temperature, typically LT = 10 K, another category of grains will also be able to record this magnetic field. In the case of magnetite, grains < 20 nm, called superparamagnetic grains (SP), will be able to record a remanent magnetization called LT-SIRM. In summary, if we assume that the ferromagnetic mineral is magnetite, the RT-SIRM is a proxy for SD and MD, whereas the LT-SIRM integrates a larger fraction with SP, SD, and MD. If a rock fragment is heated from 10 K very quickly, the LT-SIRM will lose its intensity if the fraction of SP is important. This decrease is therefore a proxy for the concentration of SP in the rock powder. This is precisely what is measured with the PM parameter: the decrease of LT-SIRM from 10 K to 35 K (Figure 1A). Here, we used the definition of Aubourg and Pozzi [9], where PM = (LT-SIRM$_{10K}$ − LT-SIRM$_{35K}$)/LT-SIRM$_{10K}$.

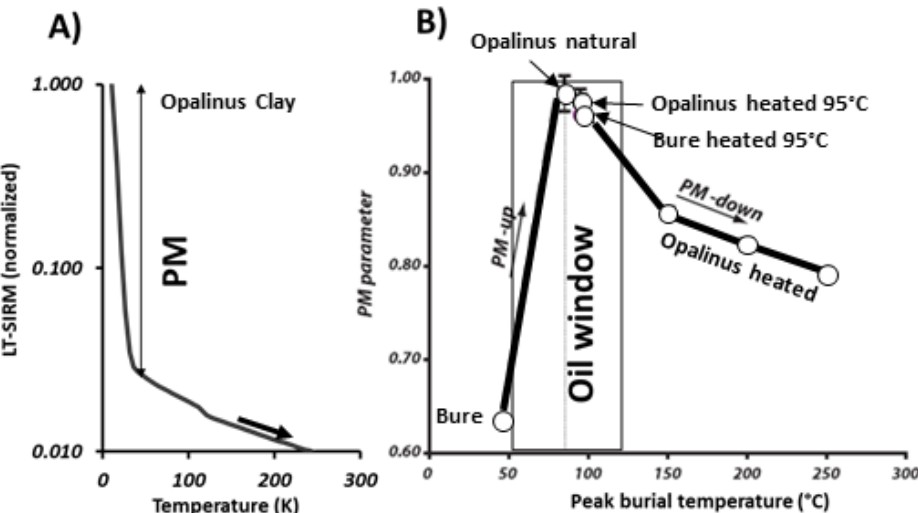

**Figure 1.** (**A**) Representative warming monitoring of artificial isothermal remanence imparted at 10 K at 2.5 Tesla (LT-SIRM) of Opalinus Clay from Mont Terri (Swiss). Note the log scale and the huge dropdown of LT-SIRM, quantified by the PM parameter. This indicated a large predominance of nanoparticles in the superparamagnetic state. (**B**) Published PM parameter versus peak burial temperature of natural shales (Opalinus and Bure) and laboratory-heated shales. The oil window is indicated. The maximum value of PM is localized in the oil window. Two branches, PM-up and PM-down, were drawn. This indicated that a PM value corresponded to two possible burial temperatures (redrawn from Aubourg and Pozzi [9].

Aubourg and Pozzi [9] studied two categories of shales, Opalinus Clay (Dogger) from the Jura mountains and Bure Clay (Callovo-Oxfordian) from the Paris Basin. The peak burial temperature of Opalinus and Bure shales has been estimated at ~85 °C and ~40 °C, respectively. They showed a PM close to 0.97 in Opalinus Clay and 0.64 in Bure Clay (Figure 1A). This indicated that: (1) the rock could contain a very large fraction of nanomet-

ric magnetic grains, and (2) this parameter could vary in significant proportions—almost 97% of LT-SIRM was reduced from 10 K to 35 K in the case of the Opalinus Clay. Aubourg and Pozzi [9] performed burial-like heating experiments using a gold capsule in an autoclave at 150 °C (53 days), 200 °C (31 days) and 250 °C (27 days). Hydrostatic pressure was maintained at 100 MPa. Based on rock magnetic analysis of natural claystones and heated claystones, Aubourg and Pozzi [9] suggested that PM displayed an evolution consistent with the peak burial temperature ($T_B$) in the range of 50 °C to 250 °C (Figure 1B). The PM evolution followed two branches: a $PM_{up}$ branch (50—90 °C) and a $PM_{down}$ branch (90–250 °C). The maximum value of PM ($PM_{max} = 0.97$) was observed for a burial temperature near ~90 °C.

This high proportion of nanograins in shales has been observed in different basins. Kars et al. [11–13] observed the diagnostic signal in samples (cores and cuttings) taken from boreholes. Abdelmalak et al. [14] reported the diagnostic signal in argillaceous rocks from volcanic margin basins (East Greenland). Chou et al. [15] reported the diagnostic signal in siltstones from a fold-and-thrust belt in Taiwan. One of the reasons for this ubiquity is that inherited magnetic iron oxides are altered during the early stage of burial in anoxic conditions [16]. This means that the essential, if not all, ferromagnetic minerals are neoformed in argillaceous rocks. A decisive experiment showed that the association of pyrite, organic matter, and a heating at 90 °C allowed the production of magnetite [17]. These components are classically present in shales.

More generally, in the context of shale burial, Aubourg et al. [18,19] proposed the existence of a window in which magnetite would be the main ferromagnetic mineral (Figure 2). The magnetite window covers a burial temperature range from ~60 °C to 330 °C [19]. At ~60 °C, magnetite can coexist with greigite. From ~340 °C, magnetite, then at a very low concentration (a few ppmv), coexists with monoclinic pyrrhotite SD and MD. Aubourg et al. [19] showed that the concentration of magnetite decreases by one order of magnitude from the oil window entrance ($T_B$~60 °C) to the metamorphic domain ($T_B$~330 °C). Above this temperature of 330 °C, pyrrhotite becomes the predominant mineral in the shales (Figure 2). Pyrrhotite is formed from magnetite and pyrite [20].

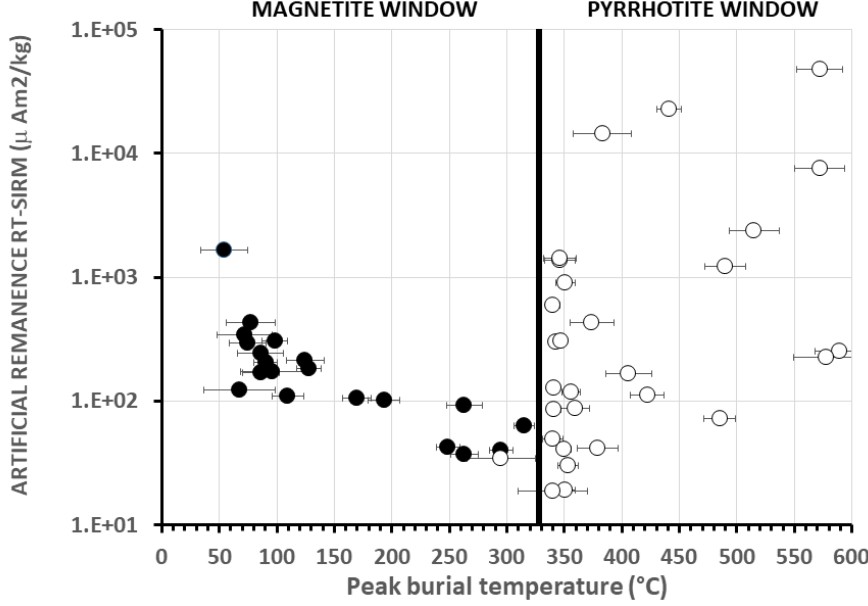

**Figure 2.** Artificial remanence (RT-SIRM) of shales from the Pyrenees and Taiwan mountain belt. Data from Aubourg et al. [19]. The black circles correspond to shales in which magnetite was the main ferromagnetic mineral. For the white circles, pyrrhotite was the main mineral. The peak burial temperature ($T_B$) of shales was derived from vitrinite analysis and from Raman spectroscopy of carboniferous material at 50 °C to ~330 °C. Pyrrhotite >1 μm was formed at the expense of magnetite and pyrite. Redrawn from Aubourg et al. [19].

## 3. Geological Setting and Sampling

In this study, we combined analysis of Borehole EST433 from the Basin of Paris [8,21] and Opalinus Clay samples from the Swiss Molasse Basin [22].

The thermal subsidence of the sedimentary basin of Paris began during the late Triassic until the early Cretaceous time [23] (Figure 3). During the Jurassic, shales and carbonates were deposited in a successive sequence of transgression and regression. Borehole EST433 was drilled in 2008, and ~1.2 km of cores of Jurassic rocks have been recovered [24]. Blaise et al. [8] examined the evolution of vitrinite reflectance within rocks from Borehole EST433. The vitrinite reflectance was evolving from 0.35% (~500 m) to 0.75% (~1900 m) (Figure 4). This indicated that the organic matter was immature to early mature in the petroleum sense. This was confirmed by a low $T_{max}$ (413–436 °C) in the RockEval analysis. The total organic matter content was generally <1%, except in the Bathonian and lower Toarcian, where content could exceed 10%. The Toarcian claystones had marine organic matter (type II kerogen) and were the only potential oil-prone source rock of the section. Otherwise, the organic matter was mainly terrestrial (type III kerogen) [21]. Peak burial temperatures derived from a set of different techniques ranged between ~40 and ~80 °C from the lower Cretaceous to the Triassic [8]. We collected 33 core samples (Figure 4, Table 1) located essentially in the Callovo-Oxfordian shales and the Lower Jurassic shales. A few months after the sampling party, we collected fragments from the cores. The rock fragments were reduced manually into powder (~500 mg) and sealed in a gel cap. Kars [25] showed that aging rock powder for few months may drastically reduce the PM parameter. For this reason, we measured the different rock magnetic parameters within a few hours after crushing.

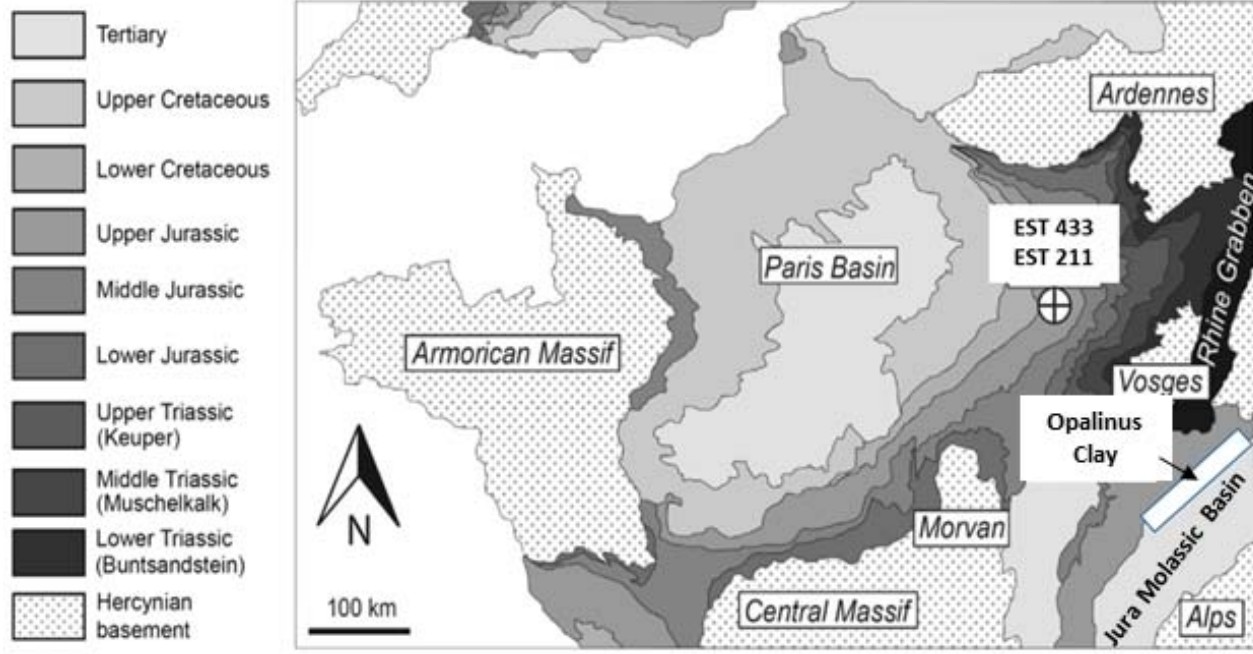

**Figure 3.** The Paris Basin, showing the locations of Boreholes EST433 and EST211. The white rectangle frames the location of boreholes where Opalinus Clay was sampled.

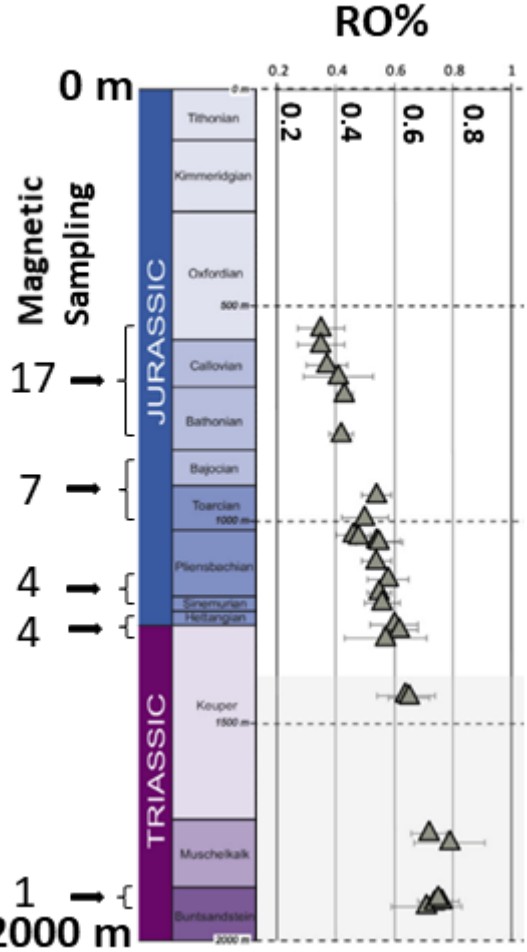

**Figure 4.** Peak burial temperature proxies from Borehole EST433. The reflectance of vitrinite (RO%) from ~0.3 to ~0.8 indicated that organic matter maturity was within the oil window. The numbers on the left refer to sampling for magnetic analysis. Data from Blaise et al. [8].

**Table 1.** Data from Borehole EST433, including depth location of the samples in the borehole. RT-SIRM: room-temperature saturated isothermal remanent magnetization; LT-SIRM: low-temperature saturated remanent magnetization (10 K). The PM value was inferred from the LT-SIRM drop between 10 K and 35 K. PM = (LT-SIRM$_{10K}$ − LT-SIRM$_{35K}$)/LT-SIRM$_{10K}$. K: low-field magnetic susceptibility; CaCO$_3$: carbonate content.

| | Depth | RT-SIRM | LT-SIRM | PM | K | CaCO$_3$ |
|---|---|---|---|---|---|---|
| | m | µAm$^2$/kg | µAm$^2$/kg | | µSI | % |
| | 531.95 | 223 | 530 | 0.66 | 24 | 83 |
| | 531.95 | 26 | 141 | 0.66 | 70 | 51 |
| | 542.10 | n.m. | 477 | 0.51 | 8 | 35 |
| | 547.95 | 127 | 733 | 0.43 | 80 | 53 |
| | 556.10 | 14 | 126 | 0.71 | 31 | 19 |
| | 566.00 | 73 | 1040 | 0.54 | 31 | 30 |
| | 573.11 | 131 | 893 | 0.47 | 84 | 25 |
| | 587.81 | 64 | 1050 | 0.59 | 115 | 26 |
| Callovo-Oxfordian | 598.00 | 84 | 1130 | 0.50 | 138 | 30 |
| | 609.25 | 78 | 1130 | 0.50 | 198 | 30 |
| | 625.04 | 82 | 1660 | 0.56 | 196 | 46 |
| | 636.85 | 48 | 1100 | 0.55 | 132 | 24 |
| | 654.29 | 133 | 945 | 0.55 | 93 | 43 |
| | 676.29 | 50 | 706 | 0.64 | 93 | 96 |
| | 725.06 | 77 | 158 | 0.27 | 12 | 100 |
| | 756.10 | 29 | 78 | 0.38 | 43 | 100 |
| | 796.50 | 7 | n.m | 0.71 | 78 | 49 |

**Table 1.** *Cont.*

|  | Depth | RT-SIRM | LT-SIRM | PM | K | CaCO$_3$ |
|---|---|---|---|---|---|---|
|  | m | μAm$^2$/kg | μAm$^2$/kg |  | μSI | % |
| Dogger | 860.14 | 25 | 570 | 0.81 | 63 | 94 |
|  | 891.97 | 15 | 71 | 0.51 | 19 | 5 |
| Toarcian | 966.38 | 58 | 2100 | 0.86 | 364 | 5 |
|  | 968.98 | 57 | 2680 | 0.92 | 331 | 43 |
|  | 1032.44 | 32 | 627 | 0.89 | 30 | 16 |
|  | 1035.55 | 26 | 538 | 0.89 | 57 | 49 |
|  | 1034.45 | 15 | 58 | 0.94 | 451 | 3 |
| Domerian Carixian | 1128.47 | 92 | 5800 | 0.93 | 257 | 3 |
|  | 1129.63 | n.m. | 7750 | 0.94 | 266 | 13 |
|  | 1186.05 | 57 | 3260 | 0.97 | 296 | 46 |
|  | 1188.40 | 57 | 3620 | 0.97 | 225 | 10 |
| Rhetian | 1247.96 | 39 | 250 | 0.61 | 75 | 11 |
|  | 1261.14 | n.m. | 68 | 0.59 | 189 | 1 |
|  | 1263.02 | n.m. | 5920 | 0.94 | 477 | 1 |
|  | 1270.95 | 322 | 314 | 0.87 | 226 | 32 |
| Trias | 1889.89 | 34 | 288 | 0.60 | 247 | 3 |

n.m. = not measured.

Opalinus Clay has long been the subject of intense investigation in the Mont Terri project, where the feasibility of geological disposal of nuclear waste is being studied [26]. Further to the east, this formation is also present at various depths in the Swiss Molasse Basin (Figures 3 and 5). Opalinus Clay is a laterally continuous, lithologically homogeneous formation with a regionally comparable diagenetic evolution (Figure 5A) [22,27]. Mazurek et al. [22] used data from apatite fission track analysis, vitrinite reflectance, and biomarker isomerization to constrain peak burial temperatures from boreholes. Peak burial temperatures were estimated from modeling and available vitrinite data (RO%). When RO% was provided, peak burial temperature was estimated from Easy%R0 thermal modeling by Mazurek et al. [22]. In addition, we proposed the empirical calibration of vitrinite data provided by Barker and Goldstein [28] (Table 2). The highest burial temperature (~115 °C) was reached eastward, for the Herdern, Berlingen, and Kreuzlingen boreholes. Lower peak burial temperature (~85 °C) was estimated for the Opalinus Clay in the Weiach and Benken boreholes. We collected fresh fragments from seven boreholes (Table 2). Rock magnetism was performed right after crushing the fresh fragments to avoid an aging effect.

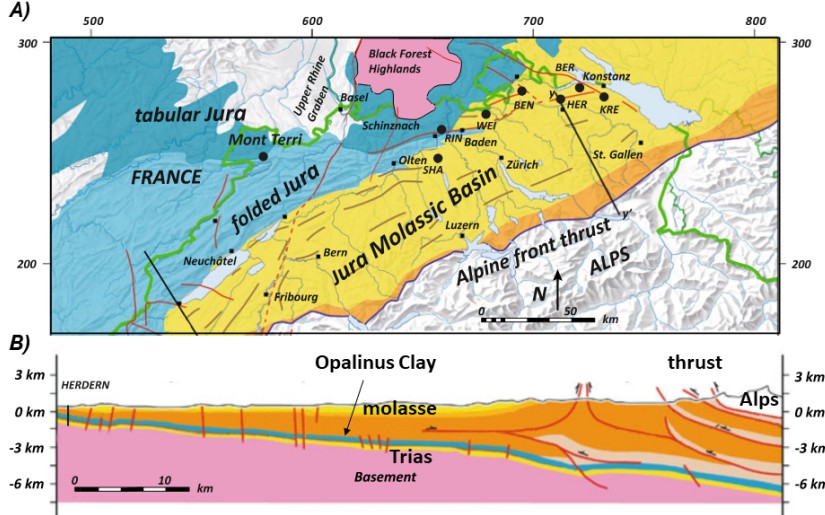

**Figure 5.** Location of boreholes from which Opalinus Clay was sampled. (**A**) Map view; coordinates are on the Swiss km grid. (**B**) Cross-section; the Herdern borehole location is indicated. Map and geological cross-section adapted from [27].

**Table 2.** Opalinus Clay data, including depth location of the samples in the borehole. RT-SIRM: room-temperature saturated isothermal remanent magnetization; LT-SIRM: low-temperature saturated remanent magnetization (10 K); K: low-field magnetic susceptibility; G%: goethite proxy derived by the slope of RT-SIRM during cooling. The PM value was inferred from the LT-SIRM drop between 10 K and 35 K. PM = (LT-SIRM$_{10K}$ − LT-SIRM$_{35K}$)/LT-SIRM$_{10K}$. RO%: percentage of vitrinite reflectance. Peak burial temperature derived from an equation by Barker and Goldstein [28]. EasyRO% and modeling are data from Mazurek et al. [22].

| Borehole | Depth | RT-SIRM | LT-SIRM | K | G% | PM | RO% | Peak Burial Temperature | | |
| | | | | | | | | B&G 89 | EasyRO% | Modeling |
| | m | µAm$^2$/kg | µAm$^2$/kg | µSI | | | % | °C | °C | °C |
| Riniken | 385.16 | 52 | 1250 | 267 | 23 | 0.83 | 0.40–0.44 | 42–54 | 60–71 | |
| Schafisheim | 1015.67 | 61 | 2010 | 234 | 25 | 0.88 | ~0.48 | ~65 | | |
| Weiach | 581.86 | 72 | 1660 | 168 | 29 | 0.78 | 0.53–0.57 | 77–86 | 87–92 | 84 |
| Benken | 635.67 | 35 | 1330 | 299 | 10 | 0.92 | 0.52–0.58 | 75–88 | 85–93 | 85 |
| Herdern | 1732.38 | 80 | 2300 | 227 | 28 | 0.86 | n.d. | | | 115 |
| Berlingen | 1912.78 | 61 | 2610 | 256 | 16 | 0.91 | n.d. | | | 110 |
| Kreuzlingen | 2175.58 | 59 | 1280 | 299 | 16 | 0.85 | n.d. | | | 115 |

n.d. = not determined.

## 4. Methods

We measured the magnetic susceptibility (MFK1 Kappabridge, Agico, Inc. Brno, Czech Republic) and monitored the remanence at low temperature (<300 K) using a Magnetic Properties Squid Magnetometer (MPMS-XL, Quantum Design, California, CA, USA). Low-temperature experiments allowed us to limit considerably the magnetic transformations, with a detection threshold on the order of one part per billion [29]. This technique is particularly suitable for detecting diagnostic magnetic phase transitions in magnetite [30], pyrrhotite [31,32], and hematite [33]. Referring to the low-temperature measurements, Kars [34] demonstrated that the internal reproducibility (same powder measured several times), as well as external reproducibility (different powders from the same rock fragment), were better than 1%. In this study, some stratigraphic levels were measured several times (different fragments taken at about few centimeters from the core) to check the consistency of the results.

For the low-temperature measurements, we monitored a remanence on cooling or warming obtained either at room temperature (300 K, RT) or at low temperature (10 K, LT). This remanence was acquired with the application of a 2.5 T magnetic field. We expected the saturation achieved and the resulting remanence was then a saturated isothermal remanent magnetization (SIRM). The LT-SIRM was monitored through the warming (10 K to 300K) in a null magnetic field (trapped field < 100 nT). The RT-SIRM was monitored on cooling under a 50 µT magnetic field. This procedure, initially proposed by Aubourg and Pozzi [9], aimed to exacerbate some magnetic transitions of antiferromagnetic minerals [35], and to illuminate an input of induced magnetization that is believed to be carried by nanosized pyrrhotite and magnetite. This is the P-behavior as defined by Kars et al. [35]. For a few specimens, we also performed the monitoring of RT-SIRM on cooling and warming. It should be noted that a few samples displayed a noisy RT-SIRM monitoring curve on cooling due to low magnetization.

We carried out scanning electronic microscopy (SEM) on selected specimens to obtain information on iron sulfides. SEM observation was performed on representative samples from Borehole EST433 (ANDRA, Bure, France). To obtain information on neoformed magnetic minerals, we performed laboratory heating at 95 °C for tens of days according to the protocol proposed by Aubourg et al. [36]. We then performed a focused ion beam (with gallium) section of a pyrite framboid. This 0.2 × 5 µm section was analyzed with a MET JEOL 2000 FX 200 Kv. Chemical analysis of sulfide and oxygen was performed using a mono-crystal analyzer (Si/Li EDX Bruker).

## 5. Results

### *5.1. Bure Samples*

#### 5.1.1. Rock Magnetism

The calcite content of the samples was obtained from ANDRA (Agence nationale pour la gestion des déchets radioactifs). It varies in large proportion, from 100% to near 1%, and can reach almost 100% in marls. By contrast, in shales, the calcite content can be as low as 1%. The low-field magnetic susceptibility (K) of our samples ranged from 8 to 477 μSI. Hysteresis loops (not shown) displayed straight lines, which indicated a dominant paramagnetic behavior. Both low magnetic susceptibility and relatively strong paramagnetic susceptibility suggested that a minute amount of ferromagnetic minerals was present in our samples. Assuming that magnetite was the main magnetic mineral, with the knowledge that the saturation remanent magnetization of magnetite is SIRM < 20 $Am^2$/kg [37], a value for RT-SIRM on the order of $10^{-4}$ $Am^2$/kg (Table 1) gives a magnetite volumetric concentration of >2 ppmv [19]. We checked for possible correlations between the calcite content, the TOC content, and the magnetic parameters. We did not observe significant correlations. However, the magnetic susceptibility tended to be lower when calcite content was elevated. The LT-SIRM was on the order of $10^{-3}$ $Am^2$/kg; i.e. one order of magnitude more than for the RT-SIRM. The ratio of LT-SIRM/RT-SIRM evolved from 1 (Rhetian sample) to 64 (Carixian sample).

Representative remanence curves of two end members are displayed in Figure 6. The depth between the two chosen samples was ~600 m. In both samples, the calcite content was lower than 30%, and the TOC was about the same (~0.6%). When cycling the RT-SIRM of the Early Oxfordian sample upon cooling and warming (Figure 6A), we noted two main behaviors: (1) near 120 K, a nonreversible Verwey transition pointed to the existence of stoichiometric magnetite (Figure 6A); and (2) upon cooling, the RT-SIRM increased regularly. When larger than ~10%, this increase can be attributed to the goethite contribution [9]. We derived from this behavior a slope that we named G%: G% = 100 × $(RT\text{-}SIRM_{1\text{-}50K}RT - SIRM_{300K})/RT\text{-}SIRM_{150K}$. For the Early Oxfordian sample, G% was 11. When warming the LT-SIRM (Figure 6B), there was a steady decrease of remanence from 10 K to 300 K. At 300 K, the residual LT-SIRM was similar to the RT-SIRM. For the Early Oxfordian sample, 40% of the LT-SIRM was lost at 35 K (PM = 0.4). This example was similar to the Callovo-Oxfordian claystones presented by Aubourg and Pozzi [9] from a different borehole (EST211) (Figure 1B).

The Carixian sample behaved differently. When cycling the RT-SIRM, the Verwey transition was much more pronounced (Figure 6C), and the parameter G% was less than 10. However, the most remarkable difference was the development of P-behavior. This P-behavior resulted from a superimposition of an induced magnetization, which paralleled the applied field of 5 μT, and the RT-SIRM [35]. Note that the Early Oxfordian sample displayed embryonic P-behavior (Figure 6A). For Carixian claystones, the LT-SIRM on warming displayed a diagnostic feature, with a PM = 0.9 and a marked Verwey transition (Figure 6D). This Carixian sample's rock magnetic signature can be compared to the Early Dogger Opalinus claystones from Aubourg and Pozzi [9]. For selected samples, the cooling of RT-SIRM and the warming of LT-SIRM are shown together (Figure 6E,F). The RT-SIRM pattern displayed P-behavior and a different goethite contribution (Figure 6E). There was a trend toward high goethite contribution and low P-behavior at low depths, and very low goethite contribution and high P-behavior at higher depths. The LT-SIRM pattern displayed a large amplitude of loss between 10 and 35 K (Figure 6F). The shallowest drop corresponded to the shallowest depth.

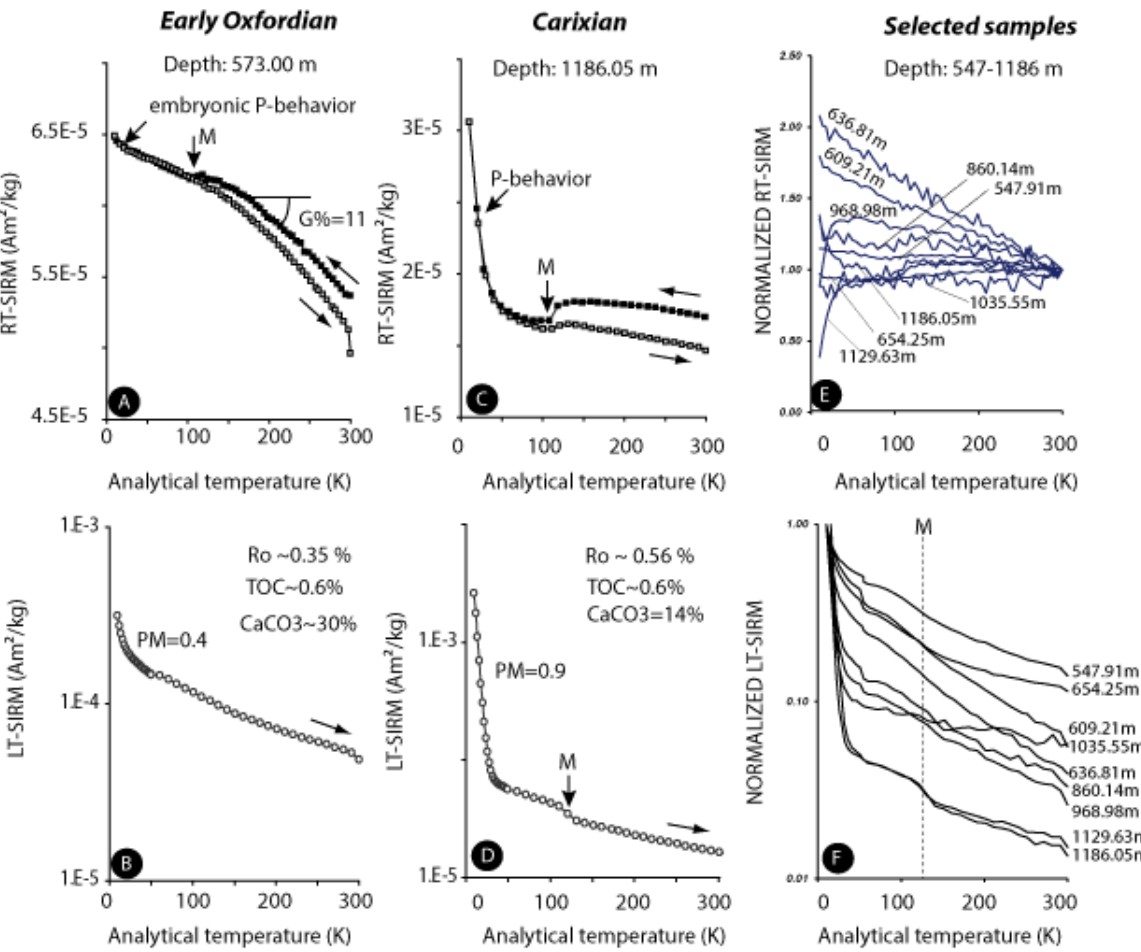

**Figure 6.** Cooling (**A**,**C**,**E**) and warming (**B**,**D**,**F**) monitoring of artificial remanence acquired at room temperature (RT-SIRM) and 10 K (LT-SIRM), respectively. All data are from Borehole EST433. Some data were published in Blaise et al. [19]. Early Oxfordian shales (**A**,**B**) displayed characteristics of immature rocks in the sense of organic matter maturity. The Carixian sample (**C**,**D**) displayed characteristics of early mature rocks. Selected samples taken at different depths (**E**,**F**); note the gradual increase of PM with depth (**F**). M is the Verwey transition at ~120 K, and was indicative of stoichiometric magnetite. The P-behavior and PM were indicative of fine magnetic particles in the superparamagnetic state.

The evolution of G% and PM are displayed in Figure 7. From 531 m to 637 m depth, G% increased from 0 to a maximum value of 60 (Figure 7A). At larger depths, G% was lower than a few %. The value of G% was not correlated to the content of calcite. The parameter PM displayed a range of 0.27 to 0.97 (Figure 7B). We distinguished three ranges of calcite content: 0–30%, 30–60%, and >60%. When samples with low calcite content (<30%) were taken, a general pattern was observed, with a quite regular increase from 0.4 at lower depths to a maximum value (0.97) at 1186 m (Figure 7B). After the maximum value of PM at ~1030 m, we noticed a sharp decrease in PM, from ~0.97 to ~0.59 at about 60 m depth. Samples fragments from the same horizon provided quite the same PM. For example, at depths of 1035.55 and 1034.45 m (lower Toarcian), five measurements were taken, and the value of PM was remarkably defined (PM = 0.91 ± 0.02). The samples with intermediate content of calcite (30–60%) displayed about the same pattern; however, calcite-rich (>60%) samples were off-trend.

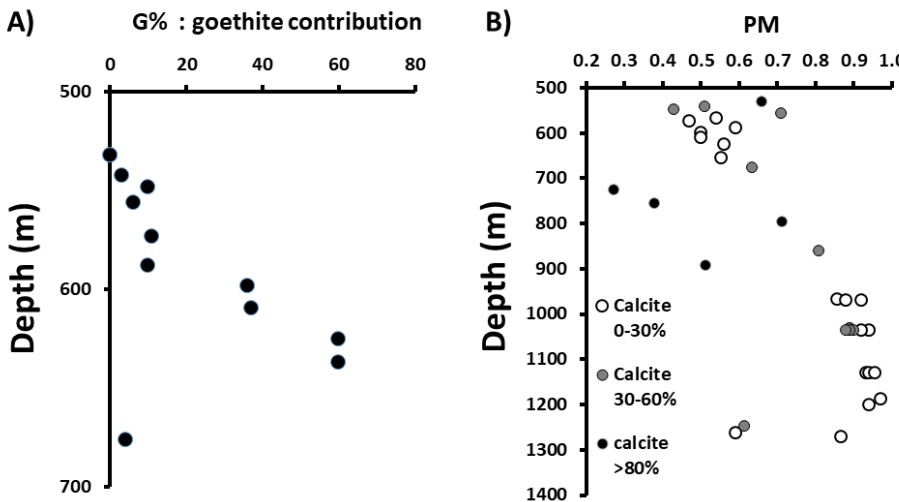

**Figure 7.** (**A**) Proxy of goethite (G%) versus depth for Borehole EST433. For depths >700 m, G% was negligible. (**B**) PM parameter versus depth; calcite content is indicated.

### 5.1.2. Microscopic Observation

We inspected fragments from Borehole EST433 using SEM, and focused our analysis on pyrite grains. Indeed, Brothers et al. [17] showed that magnetite formed at the expense of pyrite at 90 °C. It is then likely to track magnetite near pyrite framboids. The most common observation all along the borehole were framboid aggregates of euhedral pyrites of different sizes (Figure 8A). However, in Toarcian claystones (968.38 m depth), we observed some rounded grains within framboid aggregates (Figure 8B). SEM inspection of a rounded aggregate of framboids of Oxfordian claystones revealed the presence of oxidized framboids (Figure 8C), compatible with magnetite.

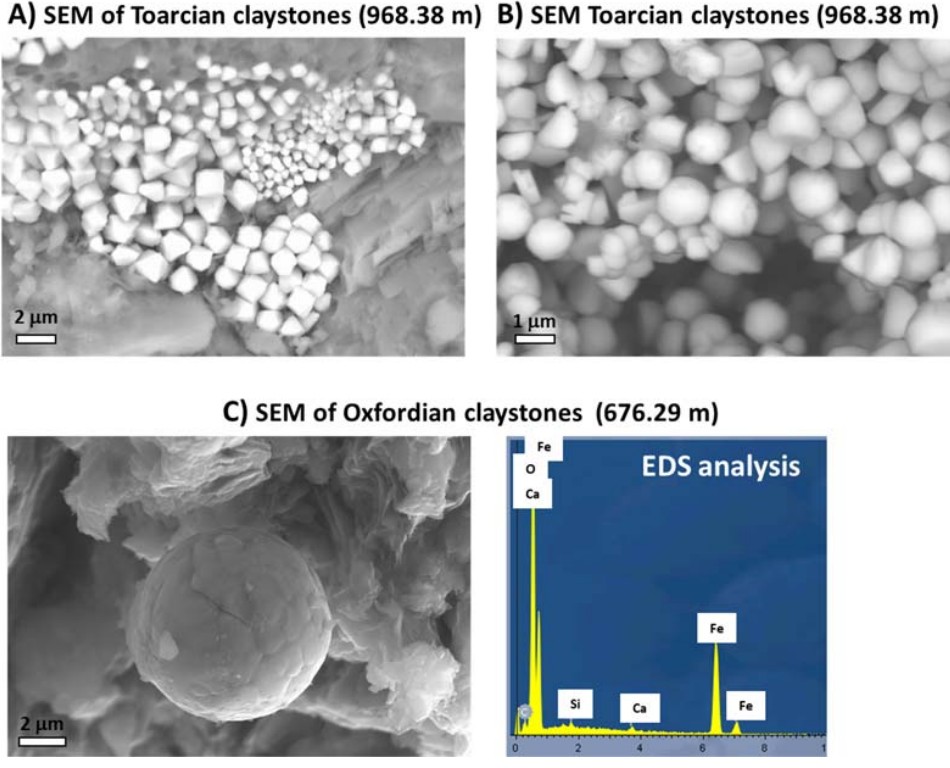

**Figure 8.** Scanning electron microscopy (SEM) observation of shales from Borehole EST433. Framboids of pyrite (**A**) and partially oxidized pyrite (**B**) observed in Toarcian claystones. (**C**) Spherule of former pyrite framboids oxidized observed in Oxfordian claystones and the EDS analysis.

To obtain more information on oxidized pyrite, we ran laboratory experiments on Callovo-Oxfordian claystones (Borehole EST211, depth 675 m). This procedure was designed to increase our chances to collect material for TEM observation. We performed laboratory heating for 67 days following the experimental protocol of Aubourg et al. [36] (atmospheric pressure, half-confined environment).

From the heated sample, we obtained a $5 \times 0.2$ μm FIB section across a framboid (Figure 9). The framboid slice showed the pyrites contoured by a gel shape and pores. To better understand the nature of this gel, we conducted a transmission electron microscopy (TEM) study coupled with EDX analysis (Figure 10). A section across two pyrite grains displayed S/Fe and O/Fe ratios (Figure 10A). The gel had an O/Fe ratio close to ~1, consistent with magnetite ($Fe_2O_3$). The pyrites had a S/Fe ratio close to 2, consistent with the formulation $FeS_2$. There was no evidence of oxidation in pyrite. When mapping sulfide (S) and oxygen (O) (Figure 10B), it was apparent that unaltered pyrite was framed by iron oxides, likely magnetite. A close inspection of the boundary of pyrite showed euhedral iron oxides of less than 20 nm in size embedded in a gel-like material (Figure 11). The nature of the gel itself remains enigmatic, and requires further investigation. Assuming that these grains encapsulated in the gel were magnetite, this size implied that they were in a superparamagnetic state (<20 nm). It is therefore plausible that these grains were responsible for the characteristic magnetic behavior (parameter PM and P-behavior).

### 5.2. Samples from the Opalinus Clay

Seven samples of Opalinus Clay were analyzed magnetically (Table 2). The magnetic susceptibility ranged from 168 to 299 μSI. Both RT-SIRM and LT-SIRM were homogenous in magnitude (average $6 \times 10^{-5}$ Am$^2$/kg and $1.7 \times 10^{-3}$ Am$^2$/kg, respectively). The ratio of LT-SIRM/RT-SIRM ranged between 22 and 43. These values were comparable to those obtained for the Opalinus Clay from Mont Terri studied by Aubourg and Pozzi [9]. G% ranged from 10 to 29, and PM ranged from 0.78 to 0.92. The RT-SIRM and LT-SIRM monitoring curves are shown in Figure 12. RT-SIRM cooling curves showed a Verwey transition at ~120 K, a potential goethite contribution, and P-behavior (Figure 12A). LT-SIRM warming curves displayed a large drop of intensity in the range of 10–35 K (Figure 12B). Those curves were similar to those identified at Mont Terri [9], and also comparable to Bure EST433 shales (Figure 6F). The PM value contrasted with PM values from Borehole EST433, which evolved from 0.40 to 0.94. We distinguished boreholes to the West (Benken, Weiach, Riniken, Schafisheim) from those to the East (Berlingen, Herdern, Kreuzlingen), which experienced a higher burial temperature [22]. Samples from Weiach and Riniken had the same magnetic signatures, with PM values of ~0.8; they also were located along an E–W trend. Further south of those two boreholes, the sample from Schafisheim developed a large P-behavior and a PM of ~0.9. Further east, the sample from Berlingen also displayed a large PM of ~0.9. South of these two boreholes, samples from the Herdern and Kreuzlingen boreholes had a PM of 0.85.

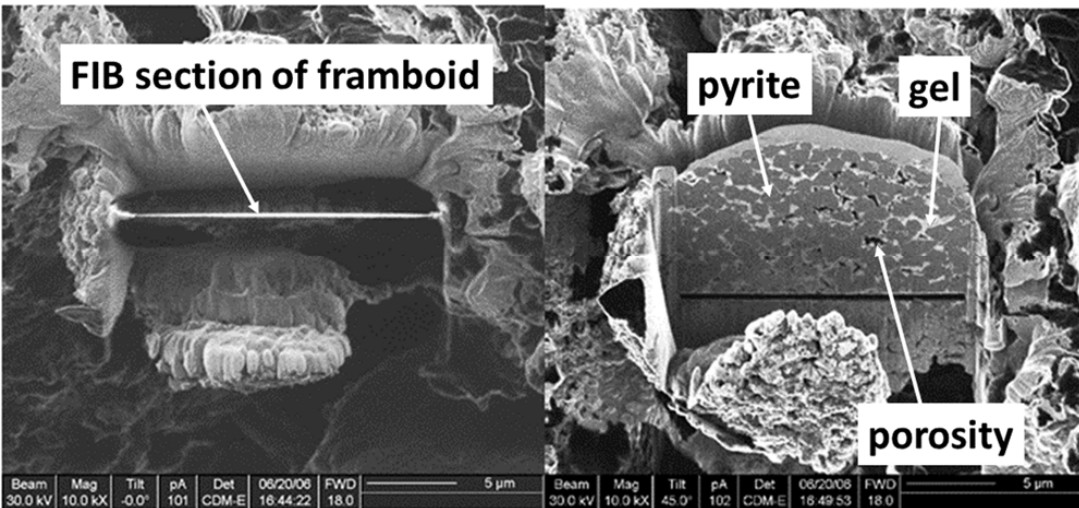

**Figure 9.** Focused ion beam (FIB) section of a pyrite framboid in Callovo-Oxfordian claystone from Borehole EST211. This sample was heated for 67 days at 95 °C. Lateral view (right) shows that pyrite (dark grey) is framed by a gel (light grey). Porosity (dark) is present within the framboids.

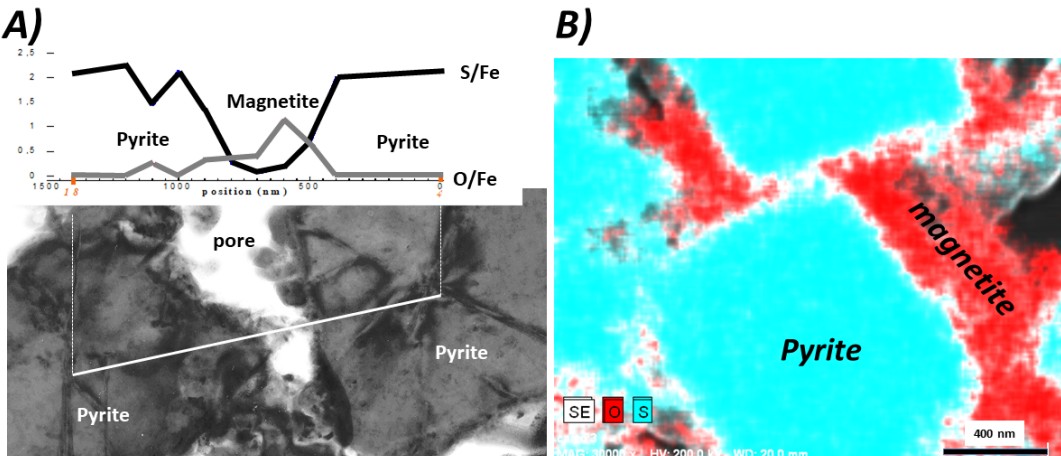

**Figure 10.** Transmitted electronic microscopy (TEM) microphotograph of pyrite from the FIB section (see Figure 9). (**A**) EDX profile between two pyrite grains. In between, iron oxide was detected. This was presumably magnetite. (**B**) EDX map of oxygen and sulfide. The pyrite grain was not oxidized, and iron oxide—presumably magnetite—framed the pyrite.

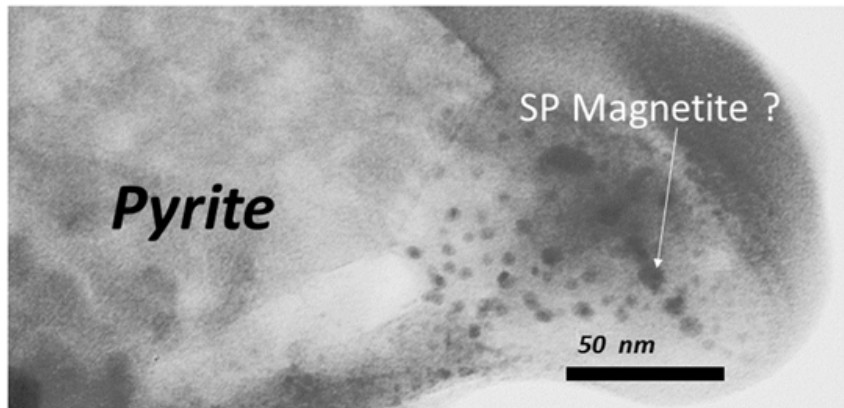

**Figure 11.** Transmitted electronic microscopy (TEM) microphotograph of pyrite framed by a gel in which nanometric euhedral crystals can be seen. These minerals were probably superparamagnetic (SP) magnetite.

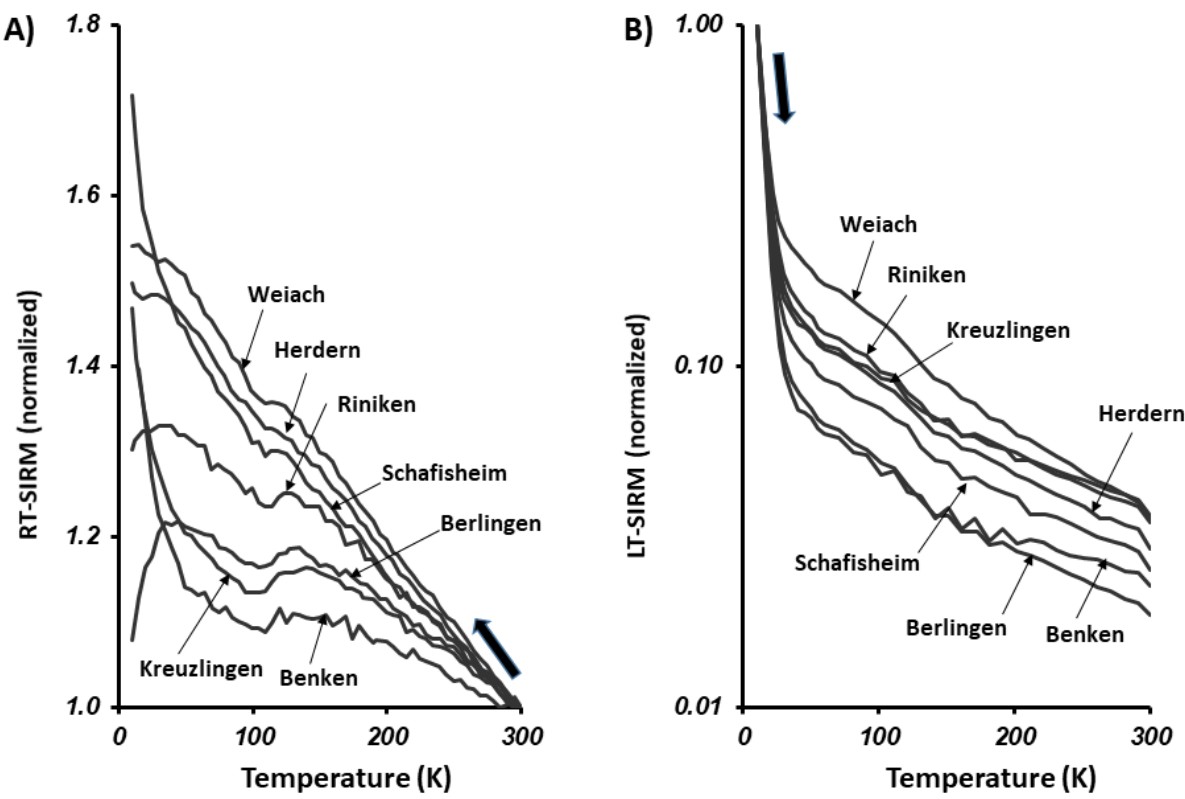

**Figure 12.** Magnetic results for Opalinus Clay. Cooling (**A**) and warming (**B**) of artificial remanence imparted at room temperature (RT-SIRM) and 10 K (LT-SIRM). The values of RT-SIRM and LT-SIRM were normalized. See Table 2 for the magnitude of the remanence.

## 6. Discussion

### 6.1. Rock Magnetism

Our rock magnetic investigation of shales from the Basin of Paris (Borehole EST433) and the Swiss Molassic Basin (Opalinus Clay) showed the presence of a magnetic assemblage constituted essentially of stoichiometric magnetite and potential goethite. Magnetite is commonly observed in moderately buried shales [18,19]. By contrast, goethite is rarely reported in shales because standard magnetic measurements at room temperature or at high temperature (>300 K) are not appropriate [25]. It should be noted that we did not detect hematite or pyrrhotite, characterized by the Morin (260 K) transition and the Besnus (34 K) transition, respectively. However, very small pyrrhotite in the superparamagnetic state was suggested by the P-behavior transition [9].

Observations of samples from the Callovo-Oxfordian at the micron- and nanoscales demonstrated that some magnetite formed at the expense of pyrite, leaving the core of the pyrite preserved from oxidation (Figures 9 and 10). The observation of pyrite grains rimmed with magnetite was in line with the laboratory experiments of Brothers et al. [17], who showed that a moderate temperature elevation of 90 °C of aqueous solutions of ferric-ligand complexes and pyrite resulted in the formation of magnetite. The replacement of pyrite with magnetite is believed to act during diagenesis. Thanks to the FIB section of a framboid previously heated at 95 °C for 67 days (Figure 10), we were able to specify that the magnetite taken in a gel form could have a size lower than 20 nm (Figure 11). We therefore proposed that in the burial processes, at least part of the magnetite in a superparamagnetic state was formed in contact with pyrites.

Aubourg and Pozzi [9] were the first to identify goethite within the Callovo-Oxfordian shales from the Bure borehole. This finding was confirmed in immature shales from Baffin Bay [14] and in shales from the Grès d'Annot formation (France) [11]. Kars et al. [25] suggested that goethite in Callovo-Oxfordian shales from Bure is not likely an alteration

by-product. Goethite occurs as nanoparticles dispersed in the clayey matrix, and not enclosed in other minerals or in organic matter. In Borehole EST433, we found goethite using the G% parameter (Figure 7A). Assuming G% is a proxy of goethite concentration, the trend of G% suggested an increase in concentration from 532 m to 637 m. This goethite window took place in the less-buried shales of the EST233 borehole. Whether this increase was due to burial processes or was related to early diagenesis remains to be demonstrated. The goethite seems to be less present for depths >650 m. Goethite occurrence also was detected in Opalinus Clay from the seven boreholes of the Swiss Molassic Basin. The G% value ranged from 10 (barely detectable) to 29 (easily detectable) (Table 2). This observation suggested that goethite occurrence was more related to rock composition rather than the burial temperature.

### 6.2. Rock Magnetism and Burial

Aubourg and Pozzi [9] and Blaise et al. [8] suggested that the PM parameter derived from low-temperature warming monitoring of artificial remanence is sensitive to burial temperature. Data from Borehole EST433 indicated that the PM was potentially erroneous when a calcite fraction greater than 60% was present (Figure 6B). Moreau and Ader [38] observed in the Paris Basin that calcite-rich samples ($CaCO_3 > 20\%$) preserved inherited magnetic minerals better than clay-rich facies. Sulfate reduction was more efficient in clayey facies, leading to greater dissolution of inherited magnetic minerals. When removing these samples with $CaCO_3 > 60\%$, the PM values displayed up and down paths, with a maximum at a depth of 1186 m. This agreed with the PM-up and PM-down branches initially proposed by Aubourg and Pozzi [9] (Figure 1B). The PM-up branch implied that magnetite formation was continuous during the diagenesis process in the studied temperature range below ~100 °C. This was suggested experimentally by Kars et al. [13] on shales heated from 50 to 130 °C. The data from Borehole EST433 thus supported the experimental work. It was the production of magnetite, probably in framboidal pyrites, that governed the PM value, as the vast majority of these magnetites were below the blocking volume (<20 nm) (Figure 11).

Comparing the EST433 data (Figure 7B) with the initial model of Aubourg and Pozzi [9] (Figure 1B), we noticed two differences: (1) the PM-up branch was not linear in shape, and had a distinguishable convex shape; and (2) the PM-down branch was much more abrupt. The convex shape of the PM-up evolution deserves some comments. It was interesting to compare this evolution to the evolution of the reflectance of the vitrinite (RO%) according to the Easy%R0 model [22,39]. Assuming a constant sedimentation rate, a constant heat flow, and a similar conductivity of rocks, the evolution of RO% should draw a convex shape. The reason for this is that the reflectance of vitrinite is a function of both temperature and time to the exposition of this temperature. That the PM-up branch displayed a convex shape has a possible profound implication regarding the mechanism of magnetic mineral formation. Similar to vitrinite, the production of magnetic nanograins is controlled by both temperature and the time of exposure to temperature. At imparted experimental temperature, Aubourg et al. [36] showed that magnetite production continued to increase over time as long as water was available in the system. These conditions must be met during burial. Kars et al. [13] complemented this observation by performing successive steps of increasing temperature. This experiment, even if very simplified compared to the natural burial mechanism, seemed to indicate that at a given temperature, a production of nanomagnetite took place, stabilized, then reactivated when the temperature increased.

Selected PM with calcite content <60% were plotted with reflectance of vitrinite data provided by Blaise et al. [8] in semilog scale (Figure 13A). Both PM and RO% varied linearly with the log of depth. PM ranged from 0.43 to 0.97, while RO% ranged from 0.35 to 0.62. PM therefore covered a wider range of values than RO% for a comparable burial. From the linear fitting curves, we could deduce an equivalent value of the reflectance of vitrinite (RO%$_C$), following the equation:

$$RO\%_C = (PM + 3.6681) \times 0.4444 - 1.5081 \tag{1}$$

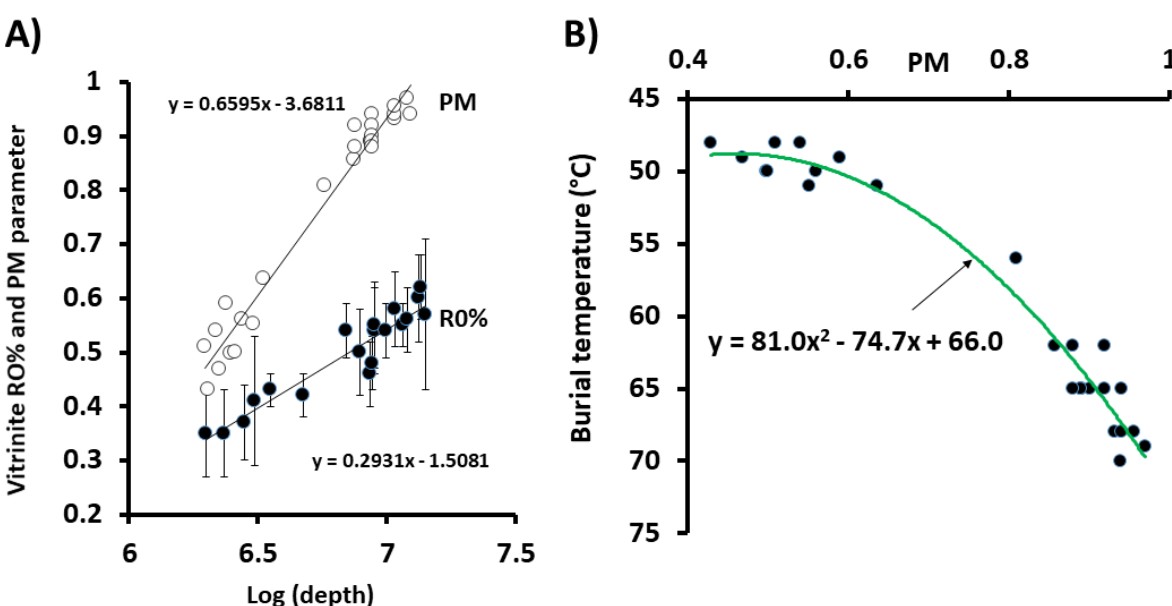

**Figure 13.** (**A**) Vitrinite reflectance data RO% (with standard deviation at one sigma) and PM magnetic parameter as a function of depth (m). Note the log scale of depth. (**B**) PM magnetic parameter for EST433 samples with $CaCO_3 < 60\%$ as a function of burial temperature, as proposed by Blaise et al. [8]. The PM instrumental uncertainty was below 1%. Best fittings were calculated for curves.

Then, PM was plotted against burial temperature estimated for Borehole EST433 (Figure 13B) [8]. The following semiempirical equation could then be established:

$$T(°C) = 81 \times PM^2 - 74.7PM + 66 \tag{2}$$

where *T* is the peak burial temperature in the range of 45–70 °C. This equation works for the entry of the so-called "oil window: of shales in the petroleum sense; i.e., the diagenesis of organic matter. We do not claim that this equation is universal. This empirical relationship was the result of a localized burial temperature model in EST433, based on the joint use of multiproxies.

It then became interesting to compare the Bure data to those from the Opalinus Clay. For this, we proposed to compare available RO% and PM values. The Opalinus Clay's PM data ranged from 0.78 to 0.92, giving vitrinite equivalents RO%$_C$ between 0.47 and 0.53 from Equation (1). From Table 2; the range of published RO% of Opalinus Clay was 0.40 to 0.58, including Mont Terri. Within the uncertainty range of vitrinite reflectance data (generally larger than 10%) and PM values (within 1% from instrumental error), we noticed that there was a good agreement between the range of RO%$_C$ deduced from PM and the measured RO%. This first-order observation showed a good agreement between the magnetic data and the vitrinite data from the Callovo-Oxfordian shales and the Opalinus Clay. The three boreholes for which the Opalinus Clay Formation reached the highest burial temperature, close to ~115°C (Herdern, Berlingen, and Kreuzlingen), had PM between 0.85 and 0.91 (Table 2). This temperature did not fit into the calibration range established with Equation (2), and these points should rather be located on the PM-down branch. However, data from EST433 suggested a rapid drop in PM after reaching a maximum (Figure 7B). Additional data are needed to better constrain this trend, as data from the Paris Basin appeared to show a much faster decrease in PM than data from boreholes in the Swiss Molasse Basin.

However, this downward trajectory implied that the magnetic nanograin population reduced beyond a burial temperature between 70 and 90 °C. This may have been either a dissolution or an increase in size above 20 nm. Kars et al. [13] showed experimentally that continuous magnetite production predominated up to an experimental temperature of

~130 °C. It should be noted that these experiments were done without pressure, and that the temperature was probably overestimated compared to natural conditions. In any case, after the magnetite production phase, most likely associated with the oxidation of pyrites, it would seem that this process stopped or even reversed. Indeed, Aubourg et al [19] observed a general decrease in the concentration of magnetite down to the metamorphic domain (Tburial > 330 °C) (Figure 2). A possible mechanism for the gradual disappearance of magnetites is their progressive replacement by pyrrhotites in the superparamagnetic state, as initially suggested by Rochette [20] and discussed by Aubourg et al. [19].

This comparison between the Opalinus Clay and the Callovo-Oxfordian from Borehole EST433 showed that it is more necessary than ever to have several markers to correctly model the burial temperature. For a burial temperature range below ~100 °C, the magnetic geothermometer has advantages over vitrinite reflectance and the RockEval technique. Beyond the nature of the organic matter, which can significantly influence the value of the proxies, it is undoubtedly the possible inheritance of detrital organic matter that can pose the most problems [40]. The organic matter of a rock in an anoxic environment can be inherited from eroded rock that has undergone higher burial temperatures than autochthonous organic matter. In the case of shales, in an anoxic environment, it was demonstrated that a large part of the detrital magnetic minerals was dissolved in the first few meters of the sediment, mainly by sulfate-reduction mechanisms [16] This dissolution of magnetic minerals during the early stages of diagenesis is of paramount importance for the use of the magnetic geothermometer. Moreau and Ader [38] observed in the Paris Basin that the dissolution of inherited magnetic minerals in carbonate facies ($CaCo_3$ > 20%) was much lower than in carbonate-poor facies, which are generally richer in organic matter. Our study confirmed this observation, since the magnetic parameters deviated from the geothermometer trends for carbonate values higher than 60%. However, it should be noted that satisfactory parameters were obtained for carbonate levels between 30 and 60% (Figure 7). One of the main flaws of the magnetic geothermometer is the existence of two solutions for the burial temperatures for an observed PM value (PM-up and PM-down curves). This is why we recommend its use for burial temperatures lower than ~100°C. The PM values can be advantageously compared to the values of RO% and $T_{max}$, the proxies of the vitrinite reflectance and RockEval.

## 7. Conclusions

This study concerns two series of shales; the shales from Borehole EST433 in the Paris Basin taken from different formations to trace the evolution of magnetic properties with depth over 1.3 km, and the Opalinus Clay taken from seven boreholes in the Swiss Molasse Basin, brought to variable burial temperatures (~85 °C to ~115 °C).

These shales showed very comparable magnetic mineral assemblages, with stoichiometric nanometric magnetites, and goethite restricted in some formations. Micrometric and nanometric observations identified nanometric magnetites around the periphery of pyrite grains in framboid aggregates. It was the concentration of magnetites smaller than 20 nm (superparamagnetic state) that dictated the evolution of magnetic properties at low temperatures (10–300 K). The PM parameter was used as a proxy to plot this evolution.

In Borehole EST433, the PM values of the shales, in which the calcite concentration was less than 60%, showed a very clear evolution with depth, with values of 0.4 to 0.9. Vitrinite values showed, in the same depth interval, values from 0.3 to 0.6%. The convex shape of PM as a function of depth reflected a temperature and time coupling effect, in a manner comparable to that of vitrinite. A calibration curve could be obtained, the first in this range of burial temperatures (~40–70 °C).

Opalinus Clay showed PMs that were consistent with published RO% values. In the warmest part of the Swiss Molasse Basin, it was likely that decreasing PM values were consistent with an evolution suggested by laboratory experiments in an anterior study.

Overall, it was noticeable that shales from two different basins, sharing similar rock magnetic signatures with PM ~0.9, and with the same vitrinite reflectance RO% in the

range of 0.5–0.6, have been the matter of different evaluations of peak burial temperatures (~70 °C in the Paris Basin, ~90 °C in the Swiss Molasse Basin).

**Author Contributions:** Conceptualization and methodology, C.A., M.K., and J.-P.P.; Microscopy: O.G.; Opalinus Samples: M.M., C.A.; writing—review and editing. All authors have read and agreed to the published version of the manuscript.

**Funding:** This research was partly funded by ANDRA (France) and grants from the Institute of Rock Magnetism (Minneapolis, MN, USA).

**Acknowledgments:** The authors wish to thank all the people who, by their advice and remarks, made it possible to identify this magnetic property of shales. The stays at the Institute of Rock Magnetism (Minneapolis, MN, USA), with the precious advice of M. Jackson, as well as the experimental work of D. Janots (ENS, Paris, France), have also favored this work. The authors would like to thank the editor O. Zhu and the two anonymous reviewers for improving this article.

**Conflicts of Interest:** The authors declare no conflict of interest. The funders had no role in the design of the study; in the collection, analyses, or interpretation of data; in the writing of the manuscript, or in the decision to publish the results.

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
