# Peer review of "A Magnetic Geothermometer in Moderately Buried Shales"

_minerals, doi:10.3390/min11090957_

Round 1

Reviewer 1 Report

Dear authors,

The publication presents interesting results and their generally good interpretation and should be published. Before that, however, I believe that it should be partially improved and that some aspects require more discussion.

My doubts concern primarily the treatment of all examined shales as almost identical mineralogically (homogeneous), assuming that their different magnetic properties are determined dominantly by factors resulting from burial heating. Meanwhile, the studied rocks represent a wide stratigraphic range and come from distant zones of the sedimentary basin - there is no doubt that sedimentary conditions (e.g. oxygenation of the seabed) were different and changed over time. This must affect the variable content of the primary ferromagnetic minerals (this is less important as they dissolve and there is little of them in relation to the secondary magnetite), but also the amount of minerals that are precursors to secondary magnetite (framboidal pyrite and maybe also phyllosilicates?).

While I believe the conclusions presented in the current version of the manuscript are generally correct, the problem described above requires more extensive discussion.

It should also be clearer describe which results were previously presented. Some of the magnetic results from drilling EST 433 have already been presented in the publication "Reconstruction of low temperature (<100 ° C) burial in sedimentary basins: A comparison of geothermometer in the intracontinental Paris Basin" Marine and Petroleum Geology, 2014. In the current version of the text it is not easy to see which results and figures are new.

Apart from this main objection, I also have a number of comments on the text. Its edition seems quite careless - it requires repeated meticulous checking by the Authors.

 My minor comments on the text are presented below:

Line 27  - I believe that the keywords list may be expanded

Line55 – Why is monodomain and not sigle-domain? The latter is used more often and will be better understood.

Line 75 – I guess 27 days

Table 1 Why the PM parameter is not shown?

Figure 5 Please explain the abbreviation SMB

I suggest that the Methods section may contain more citations from studies carried out in other laboratories.   

Line 202 – Bure samples is not easily understood. Proposes to use Paris basin.

Line 204  please explain ANDRA abbreviation

Line 279 TEM abbreviation is explained in line 284

Line 309 I suggests to use here Jura molasse basin

Line 356 For those unfamiliar with local geography and names, it won't be easy to understand what Bure borehole means

Line 379, Line 385:  Fig. 1b?

Line 425: "The 3 hottest boreholes" - I have doubts if it is linguistically correct

Yours faithfully, Reviewer

Author Response

REVIEWER 1

The publication presents interesting results and their generally good interpretation and should be published. Before that, however, I believe that it should be partially improved and that some aspects require more discussion.

My doubts concern primarily the treatment of all examined shales as almost identical mineralogically (homogeneous), assuming that their different magnetic properties are determined dominantly by factors resulting from burial heating. Meanwhile, the studied rocks represent a wide stratigraphic range and come from distant zones of the sedimentary basin - there is no doubt that sedimentary conditions (e.g. oxygenation of the seabed) were different and changed over time. This must affect the variable content of the primary ferromagnetic minerals (this is less important as they dissolve and there is little of them in relation to the secondary magnetite), but also the amount of minerals that are precursors to secondary magnetite (framboidal pyrite and maybe also phyllosilicates?).

OUR ANSWER

This is effectively quite an amazing observation that numerous observations of shales in the world display the same rock magnetic signature. This fact is explained in the introduction. We add a sentence to reinforce this observation (line 95-97 revised MS)

While I believe the conclusions presented in the current version of the manuscript are generally correct, the problem described above requires more extensive discussion.

It should also be clearer describe which results were previously presented. Some of the magnetic results from drilling EST 433 have already been presented in the publication "Reconstruction of low temperature (<100 ° C) burial in sedimentary basins: A comparison of geothermometer in the intracontinental Paris Basin" Marine and Petroleum Geology, 2014. In the current version of the text it is not easy to see which results and figures are new.

OUR ANSWER

This is right. We clarified this point in the revised MS by adding in legend of figures published data.

Apart from this main objection, I also have a number of comments on the text. Its edition seems quite careless - it requires repeated meticulous checking by the Authors.

 My minor comments on the text are presented below:

Line 27  - I believe that the keywords list may be expanded

shales, magnetite, goethite, oil window, geothermometer.

Line55 – Why is monodomain and not sigle-domain? The latter is used more often and will be better understood.

done

Line 75 – I guess 27 days

done

Table 1 Why the PM parameter is not shown?

Done. Table 1 is completed

Figure 5 Please explain the abbreviation SMB

Done.  Change in Jura Molassic Basin

I suggest that the Methods section may contain more citations from studies carried out in other laboratories.   

Done.  Add references and text line 150-154 revised MS

Line 202 – Bure samples is not easily understood. Proposes to use Paris basin.

Not taken

Line 204  please explain ANDRA abbreviation

done

Line 279 TEM abbreviation is explained in line 284

Not done. We think that it is helpful to leave the legend explanation.

Line 309 I suggests to use here Jura molasse basin

Not done

Line 356 For those unfamiliar with local geography and names, it won't be easy to understand what Bure borehole means

We think borehole is a quite standard spelling

Line 379, Line 385:  Fig. 1b?

done

Line 425: "The 3 hottest boreholes" - I have doubts if it is linguistically correct

Done “The 3 boreholes for which the Opalinus Clay Formation reached the highest burial temperature close to ~115°C”

Reviewer 2 Report

This paper presents comparable rock magnetic data from relatively shallow depths and burial temperatures from boreholes in the Paris  Basin and the Jura molasse basin. They report rock magnetic data which indicates the presence of magnetite nanograins (PM) which they argue correlates to vitrinite reflectance (maturity) values of the organic matter in the cores. They suggest that PM can be used as a geothermometer in shales with < 60% CaCO3.

The authors provide a useful review of their previous work as well as a clear and concise description of how they determined PM.  

The SEM/TEM data and pics are very useful and their hypotheses regarding mineralogy are very reasonable. I would like to know more about how they identified the gel they describe. They should label the gel on Fig. 9. How do they know that the apparent SP magnetite in Fig. 11 is in a gel?

The rock magnetic interpretations of the RT-SIRM and LT-SIRM are reasonable.

I have some questions about the PM% versus calcite % data (Figure 7). The less than 30% calcite data has one group of data points around 600m and .5 PM and then a second group of points at 1000-1200 m with a PM that shows a slight increase from ~8.5 PM at 1000m to 9 PM at 1200 PM. There are no data points for <30% CaCO3 between 650-1000m. The less than 30% calcite data looks like two sets of data points and the trend is not convincing. The 30-60% calcite shows an increase from 500 m to 1000 m  but there are only 2 data points that define the trend.   It might be better if the authors focused on noting that the <60% calcite data points define a possible trend rather than splitting the data into the <30 and 30-60 groups.   

They suggest goethite occurs dispersed “as nanoparticles dispersed in the clayey matrix.” Although not crucial for this paper, do they think the goethite occurs on clay particle surfaces?

L418-432 – I assume they did not show a figure of PM versus vitrinite from the Opalinus Clay rather than just refer to table 2 because there are only four points? If I am reading this correctly in Table 2, Ro increases from  0.40-0.44 to 0.52-0.58 while PM increases from .83 to .92. But Weiach has a relatively high Ro (0.53-.0.57) but a lower PM (.78) than the two cores with lower Ro. This raises questions about the validity of the PM-Ro trend in the Opalinus Clay. They clearly need more data from the Opalinus Clay as the author’s state. Perhaps they should deemphasize the trend and state that more data is needed to determine if a trend is present. I agree that the shales from Paris Basin and the Opalinus Clay have similar rock signatures, but I am not as convinced by the trend in the Opalinus Clay. 

I also wonder about the errors around the PM and Ro data. They mention that the uncertainty range of

Vitrinite Reflectance and PM data on line 422 but I did not see where they defined or discussed the potential errors. They should discuss this.

There are a few minor comments on the manuscript. 

Although more data would strengthen the manuscript, I think there is a relationship between PM and maturity. I recommend publication after moderate revision assuming the comments above are addressed. 

Author Response

REVIEWER 2

This paper presents comparable rock magnetic data from relatively shallow depths and burial temperatures from boreholes in the Paris  Basin and the Jura molasse basin. They report rock magnetic data which indicates the presence of magnetite nanograins (PM) which they argue correlates to vitrinite reflectance (maturity) values of the organic matter in the cores. They suggest that PM can be used as a geothermometer in shales with < 60% CaCO3.

The authors provide a useful review of their previous work as well as a clear and concise description of how they determined PM.  

The SEM/TEM data and pics are very useful and their hypotheses regarding mineralogy are very reasonable. I would like to know more about how they identified the gel they describe.

OUR ANSWER

We have not done XRD analysis on the gel. We add a sentence in the revised version to clarify this.

They should label the gel on Fig. 9. How do they know that the apparent SP magnetite in Fig. 11 is in a gel?

OUR ANSWER

This is a statement, as quoted by interrogation mark in the figure and text.

The rock magnetic interpretations of the RT-SIRM and LT-SIRM are reasonable.

I have some questions about the PM% versus calcite % data (Figure 7). The less than 30% calcite data has one group of data points around 600m and .5 PM and then a second group of points at 1000-1200 m with a PM that shows a slight increase from ~8.5 PM at 1000m to 9 PM at 1200 PM. There are no data points for <30% CaCO3 between 650-1000m. The less than 30% calcite data looks like two sets of data points and the trend is not convincing. The 30-60% calcite shows an increase from 500 m to 1000 m  but there are only 2 data points that define the trend.   It might be better if the authors focused on noting that the <60% calcite data points define a possible trend rather than splitting the data into the <30 and 30-60 groups.   

OUR ANSWER

This is a correct observation. However, we find useful to distinguish between these groups.

They suggest goethite occurs dispersed “as nanoparticles dispersed in the clayey matrix.” Although not crucial for this paper, do they think the goethite occurs on clay particle surfaces?

OUR ANSWER

We have no direct evidence for the localization of goethite. The occurrence of goethite is still a matter of debate (see Kars et al., 2015) and it is generally not observed. This is why we keep the goethite identification in this MS, because it is likely a diagenesis product.

L418-432 – I assume they did not show a figure of PM versus vitrinite from the Opalinus Clay rather than just refer to table 2 because there are only four points? If I am reading this correctly in Table 2, Ro increases from  0.40-0.44 to 0.52-0.58 while PM increases from .83 to .92. But Weiach has a relatively high Ro (0.53-.0.57) but a lower PM (.78) than the two cores with lower Ro. This raises questions about the validity of the PM-Ro trend in the Opalinus Clay. They clearly need more data from the Opalinus Clay as the author’s state. Perhaps they should deemphasize the trend and state that more data is needed to determine if a trend is present. I agree that the shales from Paris Basin and the Opalinus Clay have similar rock signatures, but I am not as convinced by the trend in the Opalinus Clay. 

OUR ANSWER

Yes, we decided that plotting 4 data is useless. It should be emphasized that RO% has quite a large uncertainty. This is why we opted for a qualitative analysis.

I also wonder about the errors around the PM and Ro data. They mention that the uncertainty range of

OUR ANSWER

We discussed about uncertainty of PM. Error bar of RO% are shown. The subject of uncertainty in vitrinite data is not sufficiently addressed in the literature. How are averages calculated, for example, when there is a bimodal distribution, one controlled by diagenesis, the other by inheritance? This is a real subject, but one for the specialists in the field.

Vitrinite Reflectance and PM data on line 422 but I did not see where they defined or discussed the potential errors. They should discuss this.

OUR ANSWER

“Within the uncertainty range of vitrinite reflectance data (generally larger than 10%) and PM values (within 1% from instrumental error)”

There are a few minor comments on the manuscript. 

Although more data would strengthen the manuscript, I think there is a relationship between PM and maturity. I recommend publication after moderate revision assuming the comments above are addressed. 

Round 2

Reviewer 2 Report

Considering the importance of the PM% versus calcite % data (Figure 7), I would at least address the observation about the less than 30% calcite and 30-60% calcite versus %PM.